

# LEET: stock market forecast with long-term emotional change enhanced temporal model

Honglin Liao[1], Jiacheng Huang[1] and Yong Tang[2]

[1] Maynooth International Engineering College, Fuzhou University, Fuzhou, Fujian Province, China
[2] School of Economics and Management, Fuzhou University, Fuzhou, Fujian Province, China

Corresponding authors
Honglin Liao,
onglinguge@gmail.com
Yong Tang, tangyong2018@126.com

## ABSTRACT

The stock market serves as a macroeconomic indicator, and stock price forecasting aids investors in analysing market trends and industry dynamics. Several deep learning network models have been proposed and extensively applied for stock price prediction and trading scenarios in recent times. Although numerous studies have indicated a significant correlation between market sentiment and stock prices, the majority of stock price predictions rely solely on historical indicator data, with minimal effort to incorporate sentiment analysis into stock price forecasting. Additionally, many deep learning models struggle with handling the long-distance dependencies of large datasets. This can cause them to overlook unexpected stock price fluctuations that may arise from long-term market sentiment, making it challenging to effectively utilise long-term market sentiment information. To address the aforementioned issues, this investigation suggests implementing a new technique called Long-term Sentiment Change Enhanced Temporal Analysis (LEET) which effectively incorporates long-term market sentiment and enhances the precision of stock price forecasts. The LEET method proposes two market sentiment index estimation methods: Exponential Weighted Sentiment Analysis (EWSA) and Weighted Average Sentiment Analysis (WASA). These methods are utilized to extract the market sentiment index. Additionally, the study proposes a Transformer architecture based on ProbAttention with rotational position encoding for enhanced positional information capture of long-term emotions. The LEET methodology underwent validation using the Standard & Poor's 500 (SP500) and FTSE 100 indices. These indices accurately reflect the state of the US and UK equity markets, respectively. The experimental results obtained from a genuine dataset demonstrate that this method is superior to the majority of deep learning network architectures when it comes to predicting stock prices.

# INTRODUCTION

The stock market is a macroeconomic indicator that reflects the shared expectations of investors regarding future market conditions. Furthermore, the correlation between stock market indices and other economic indicators indicates that stock price predictions are

crucial in comprehending and forecasting economic shifts. Forecasting stock prices can offer investors valuable insights into market and economic trends, enabling them to make informed decisions. Additionally, such forecasting can serve as an important reference point for governments as they formulate and adjust macroeconomic policies.

The prediction of stock prices has a lengthy history, with efficient capital market theory (*Fama, 1970*) putting forward the efficient market hypothesis. This hypothesis assumes that, within a legally sound, well-functioning, transparent, and competitive stock market, rational investors are capable of promptly and rationally responding to all market information, and asset prices thus reflect all obtainable information. Therefore, stock prices will reflect all material facts, including the current and future value of a company, accurately, adequately, and promptly. Nonetheless, other researchers have questioned the theory due to its excessively rigid assumptions. Throughout the long history of the capital market, numerous forecasting methods have been suggested, such as technical analysis, fundamental analysis, and time series analysis (*Murphy, 1999*). With the advancement of science and technology, various models for forecasting stock prices have been developed. The majority of traditional methods for predicting stock prices use statistical time series models like autoregressive moving average (ARIMA) and generalized autoregressive conditional heteroskedasticity (GARCH) (*Hu et al., 2020*). However, the aforementioned models necessitate manual parameter selection, which is more subjective and data-heavy, whereas statistical-based time series models employ linear models to fit stock prices. The latter approach proves challenging when characterizing multiple inter-relational variables and capturing an extensive amount of volatility and non-linear information. Furthermore, it fails to precisely model non-linear relationships, thus resulting in lower prediction accuracy.

Real-life economic, social, and irrational human behaviors can impact investors' emotions, mental states, and behaviors. This can result in share prices deviating from the model's predicted fluctuations. For instance, psychological factors like herd mentality can cause sudden stock market fluctuations with a single news story. Whilst numerous studies have outlined (*Chen et al., 2022*; *Li, Fu & Zheng, 2017*) a notable correlation between market sentiment and stock prices, only a limited number have explored the incorporation of sentiment analysis for stock price forecasting. Major political and economic events across the globe—such as the UK's departure from the European Union (EU)—have resulted in significant changes to the economic and monetary policies between the UK and the EU. As a result, these changes have produced great levels of uncertainty and challenges for the UK and the EU's economic systems, financial markets, and business operations, bringing about a long-term and far-reaching impact on the stock market (*Qiao et al., 2021*). Simultaneously, the momentum effect of the stock price (*Li, 2020*) prompts optimistic market sentiment whereby investors perceive the stock as a more attractive investment opportunity and consequently opt to purchase it, resulting in further escalation of the stock price. Similarly, when the stock price drops, market sentiment becomes more pessimistic. Investors may perceive the stock as risky and decide to sell, further reducing its value. Consequently, analyzing the long-term correlation between market sentiment and

stock price can support investors in making more informed predictions and developing effective investment approaches to mitigate risk and increase returns.

Recent advancements in deep learning techniques have led to the creation of new and effective models for time series forecasting. These models possess the capability to grasp complex non-linear interrelationships and capture non-linear fluctuations in stock prices by employing various technical strategies such as multi-layer architectures, non-linear activation functions, large data sets, and optimization of parameters. An increasing number of researchers endeavour to employ deep learning algorithms for stock price prediction to reduce human biases (*Shahi et al., 2020*; *Bhandari et al., 2022*; *Chung & Shin, 2018*; *Zou et al., 2023*).

Nevertheless, conventional time-series neural networks (incorporating recurrent neural networks (RNNs) and long short-term memory (LSTMs)) exhibit restricted memory units. Despite being equipped to handle time-series data, they primarily concentrate on short-term dependencies and are inadequate in capturing long-term dependencies. Long-term considerations, such as macroeconomic conditions and industry trends, are paramount in stock price forecasting. Nonetheless, traditional time series neural networks, such as RNNs and LSTMs, have limited storage capacity, resulting in them typically focusing on short-term dependencies and inability to capture long-term dependencies (*Vaswani et al., 2023*). Traditional time series neural networks adopt a fixed length sliding window for feature extraction. This approach hinders the combined utilization of information from different time scales in the time series. Consequently, this approach faces challenges in dealing with long-distance dependencies in huge datasets and gives minimal consideration to the global context. This ultimately weakens the ability of the model to capture overall trends and produce accurate forecasts.

To address the challenge presented by deep learning architectures in handling extensive datasets with long-distance dependencies, as well as the limitations in utilizing data at various time intervals, and to integrate market sentiment into model forecasts, this article introduces the Long-term Emotional Change Enhanced Temporal Analysis (LEET) method. LEET effectively extracts long-term dependencies of both market sentiment and stock fundamental indicators. LEET methodology presents two estimation techniques Exponential Weighted Sentiment Analysis (EWSA) and Weighted Average Sentiment Analysis (WASA), to derive market sentiment indices. Additionally, the study proposes a Transformer architecture based on ProbAttention with rotational position encoding for enhanced positional information capture of long-term emotions to effectively extract the long-term dependence of market sentiment and stock fundamentals. Based on experimental findings, this approach is effective in achieving accurate stock price predictions.

The article is structured as follows: Consistent citation and clear formatting are also maintained to ensure the academic integrity of the article. "Related Works" provides an overview of the related literature; "Methodology" explains the design and methodology; "Experiment" presents the experimental setup and findings; and finally, "Conclusion" outlines the conclusions. The language used throughout the article is formal and objective in order to ensure clarity and accuracy. Where necessary, technical terms are defined and

abbreviations explained. The vocabulary used adheres to British spelling conventions and grammatical rules. The author has made an effort to remain unbiased and balanced in their writing, avoiding evaluative language and subjective opinions.

## RELATED WORKS

### Stock prediction based on deep learning for time series

Due to the high complexity of financial markets, the combination of deep learning techniques with financial time series forecasting is considered to be one of the most attractive topics. For example, LSTM, as an improved version of traditional recurrent neural networks (RNN), has stronger memory and the ability to prevent gradient explosion. *Shahi et al. (2020)* compared the performance of LSTM and gated recurrent unit (GRU) models in stock market prediction under the same conditions, and also found that combining financial news sentiment and stock features as inputs can significantly improve the performance of stock price prediction (*Bhandari et al., 2022*) predicted the closing price of the S&P 500 index using an LSTM model. The researchers compared the efficacy of single-layer and multi-layer LSTM models and found that the former outperformed the latter in a majority of the configurations. *Chung & Shin (2018)* utilized a hybrid methodology that combined a LSTM network with a genetic algorithm (GA) to determine the optimal time window size and topology for the LSTM network. The findings are reported in their study. Compared to the benchmark model, the prediction error mean squared error (MSE) is reduced by 13.11%, while the mean absolute error (MAE) and mean absolute percentage error (MAPE) are both reduced by 12.80% and 0.19%, respectively. *Zou et al. (2023)* provide a comprehensive and up-to-date survey of stock market forecasting techniques, filling the gap between the extensive coverage of traditional methods and older neural network techniques and the lack of coverage of more recent techniques in previous surveys. These improvements demonstrate the superiority of the Transformer model proposed by *Vaswani et al. (2023)*, which incorporates a self-attention mechanism that can capture long-distance dependencies and efficiently perform temporal modeling. *Wang et al. (2022)* employed an encoder-decoder architecture and the Transformer model to forecast the CSI 300, S&P 500, Hang Seng, and Nikkei 225 indices. They compared this approach with a conventional deep learning model and found that the Transformer model had superior predictive accuracy in all experiments and a better NAV profile. *Li et al. (2022)* proposed a stock movement prediction model based on a Transformer and attention mechanism, and used social media text and stock price data including Twitter for experimental validation, by deeply extracting features of text and stock price, and using attention mechanism to obtain key information. The experimental results show that the method outperforms other baseline models in several metrics and has practical application value. *Luo, Zhuo & Xu (2023)* solved the problem of large market price fluctuations by proposing a deep neural network-based financial risk prediction model that analyses the influencing factors of carbon emission right price from multiple perspectives and effectively predicts the financial risk in the carbon trading market. *Liu et al. (2024)* proposed a new time series prediction model, LSTNet-Prophet, which, by organically combining deep learning and machine learning and tuning with particle swarm

optimization algorithm, achieves better capture of long-term dependencies in time series data and handles cyclic changes in trend, thus solving the problem of traditional methods in predicting nonlinear and nonstationary time series data. The problems of traditional methods in forecasting non-linear and unstable time series data are better solved. *Yin et al. (2023)* proposed a method called Stock Sector Trend Prediction (SSTP) for solving the problem of stock sector trend prediction. The method uses relative price strength (RPS) and temporal spatial network (TSN) to extract the temporal features of stock prices and is validated using data from stock sector indices of China's A-share market. The experimental results show that taking the stock industry as the prediction object is more stable and feasible compared to the prediction of individual stocks. Meanwhile, by constructing a multi-scale RPS time series and stock industry relationship map, the trend of the stock industry can be better described. *Zhou et al. (2021)* introduced a proficient Transformer model named Informer, which significantly improves the inference speed of long sequence prediction compared to the conventional Transformer architecture, thanks to the implementation of self-attentive distillation operation and generative decoder mechanisms. The position encoding employed by the traditional Informer is enhanced through the use of sine-cosine functions, which screen the inner product of the key and value. However, this method fails to take into account the difference in distance between the key and value, making it impossible to determine their exact proximity. This research enhances the positional embedding of ProbAttention by rotating the positional encoding. This enables the model to calculate the inner product of the two in attention, resulting in a sequential model that gains a better understanding of long-dependent emotions. The aforementioned prevalent deep learning models will be meticulously compared in the experiments.

## Stock price changes analysis based on sentiment

Researchers analyzed online opinion data to more accurately predict stock market prices. *Chen et al. (2022)* used the unsupervised learning latent Dirichlet allocation (LDA) topic model to analyze the impact of investor sentiment on stock prices. They found that investor sentiment by topic was positively correlated with excess returns, with different topics having varying impacts on stock prices over time. *Li, Fu & Zheng (2017)* analysed the impact of investor sentiment on stock prices in the Shanghai Stock Exchange forums, using a Hidden Markov Model based on the Hidden Markov Model. and modeled the relationship between sentiment indices and stock price movements through multiple regression analysis. *Das et al. (2022)* employed a logistic regression algorithmic approach for sentiment analysis and stock price prediction based on 1,452 comments retrieved from the official website of the Oromo Democratic Party, achieving a 98.32% prediction accuracy. Meanwhile, *Weng, Lin & Zhao (2022)* utilised a finely-tuned BERT model to compute investor sentiment indices of stock pre-openings and merged them with the underlying stock ticker data. Finally, the LSTM algorithm is employed to anticipate the closing costs of Ping An Bank (000001), ZTE (000063), and Muyuan (002714) of the Chinese A-share market. The experimental outcomes exhibit that the integration of the investment sentiment index enhances the predictive accuracy of the LSTM algorithm

significantly. *Shilpa & Shambhavi (2023)* used deep learning classifiers for stock market prediction and proposed a new SIWOA model and a deep confidence network (DBN) integrating stock prices and news sentiment in order to solve the problems of dependency, long execution time, and poor reliability in stock market prediction. *Ko & Chang (2021)* used a natural language processing tool, BERT, to identify the sentiment of text and applied a long short-term memory neural network (LSTM) to make stock price predictions using historical stock trading information and text sentiment. Experiments demonstrate a 12.05% accuracy improvement in the root mean square error (RMSE) of their proposed model. *Huang et al. (2021)* used long and short-term memory networks for text mining and cognitive network analysis to map their changes, and for the first time investigated and revealed in detail the evolution of emotions (positive, negative, *etc.*) with the five stages of the learning process (initial stage, conflict sublimation and stabilisation) under different levels of interactions (surface, deep and socio-emotional) in blended learning. It was found that as learning progressed, the links between the different emotions and the deeper interactions changed. Compared to the subjective report or manual coding research method, it provides a new and more systematic and dynamic analytical perspective for studying emotions and interactions in blended learning. *Xiao et al. (2023)* proposed the CoolNet model to better address the multimodal aspect-based sentiment analysis task. CoolNet utilises visual GNN to segment images into object-level graph structures to achieve detailed extraction of visual features of images; at the same time, it utilises syntactic dependency trees to capture syntactic knowledge from text and designs graph convolutional networks to achieve multimodal fine-grained alignment of semantic and syntactic levels. It also uses grammatical dependency trees to capture grammatical knowledge from text, and designs graph convolutional networks to achieve multimodal fine-grained alignment at semantic and syntactic levels. Through these two improvements, the CoolNet model effectively solves and exploits the rich information from different modalities, and achieves a deeper fusion of multimodal features at different levels. *Ding et al. (2023)* integrate Airbnb listing type and price data into the text analysis of guest reviews through structural theme modelling, and compare the frequency of occurrence of the extracted topics under different listing types (full and shared listings) and different price levels, and identify for the first time the major differences in the service attributes and needs that guests focus on in their online reviews under these influencing factors, which provides opportunities for Airbnb hosts to make customised pricing and service strategy recommendations. Meanwhile, we demonstrate the advantages of structural theme models in analysing the implicit information in this type of user review text data, which can serve as a reference for subsequent research. *Gong et al. (2023)* proposed a multi-stage hierarchical relational graph neural network to address the issue of ignoring modal imbalance contribution and relational interaction in multimodal sentiment analysis. The proposed method dynamically adjusts modal contribution and learns different types of inter-modal interactions separately.

This literature review mirrors the goal of the research described in the previous section, which was to recommend LEET as a solution to the challenge of making effective forecasts on long-term time scales and the difficulty of taking into account the complexity of long-

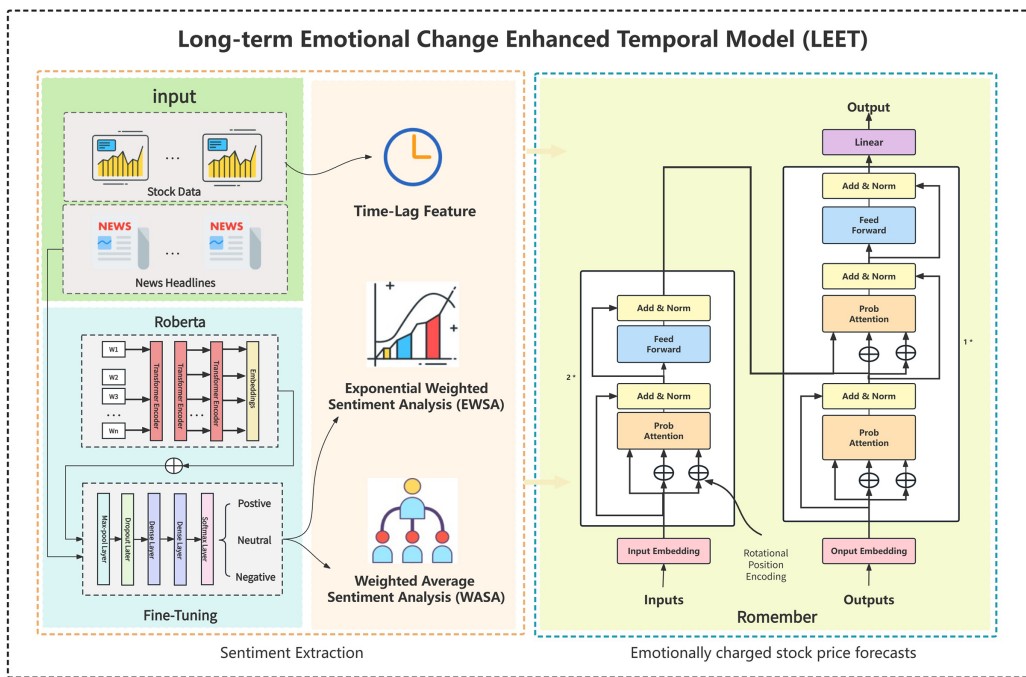

**Figure 1 Overall processing framework.** The picture illustrates the overall framework of the model. From left to right, the sections represent the extraction of sentiment value from daily news headlines, two sentiment information features EWSA and WASA, and Romember, a stock price forecasting model that captures long-term sentiment value.               

term market sentiment in the forecasting results. At the same time, this literature review echoes the objectives of the research described in the previous section, which aimed to propose LEET as a solution to the challenges of making effective forecasts on long-term time scales and the difficulties of reflecting long-term market sentiment in the forecasting results.

# METHODOLOGY

## Overall processing framework

In this section, the authors will elaborate on two core topics: sentiment extraction and temporal modeling.

Figure 1 illustrates the overall framework of the model. From left to right, the sections represent the extraction of sentiment value from daily news headlines, two sentiment information features EWSA and WASA, and Romember, a stock price forecasting model that captures long-term sentiment value.

## Problem definition

In the issue of stock forecasting, the method's input comprises the stock price and sentiment information at the end of the stock market on day L, and the output is the closing stock price on day T in the future.

$X = (x_1, x_2, \ldots, x_L),\ X \in R^L$ denotes the time series of stock prices in the input, where each term $x_i$ represents the stock price on day i of the series. The sentiment information

sequence input can be denoted as $E = (e_1, e_2, \ldots, e_L), E \in R^L$, where $e_i$ represents the sentiment characteristics on day i in the sequence. The predicted stock price value on day i can be denoted as $Y = (y_{L+1}, y_{L+2}, \ldots, y_{L+T}), Y \in R^T$, where $y_i$ is the output. Typically, the case T = 1 is considered due to the instantaneous nature of the sentiment's impact on stock prices.

## Sentiment extraction

In this section, we will explore techniques for handling vast amounts of news text data and arranging the associated sentiment information. This aligns with the "Sentiment Extraction" section depicted in Fig. 1 and provides additional insight into the origin of the sentiment data $e_i$ in the model inputs. On day i, a multitude of news headlines $News_i = \{News_i^1, News_i^2, \ldots, News_i^{M_i}\}$ can be acquired from the media. Mi denotes the number of headlines which are collected on day i. In order to obtain the sentiment tendency feature of the current news article, the pre-finetune text model for sentiment analysis, RoBERTa, is utilised to get the sentiment tendency feature of the current news $Sent_i^m$ for each article headlines $News_i^m$. The values of $Sent_i^m$ represent positive, neutral or negative sentiment and are denoted by −1, 0 and 1, respectively. The emotional profile of each news article can be specifically represented as follows:

$$Sent_i^m = RoBERTa(News_i^m), \ Sent_i^m \in \{-1, \ 0, \ 1\} \tag{1}$$

To express the sentiment profile of each news article, we need to aggregate the sentiment values of the day. Thus, the sentiment feature of the day can be stated as follows:

$$Sent_i = \frac{1}{M} \sum_{m=1}^{M} Sent_i^m \tag{2}$$

Though the stock market is closed, market sentiment is generated daily and influences the stock price on the opening day of trading. Sentiment also affects market prices over time, so it is necessary to calculate stock market sentiment during the previous day's time window. Assuming the window size is W, which is a hyperparameter we can adjust, we can obtain the market sentiment during the window period by averaging. This is referred to as Weighted Average Sentiment Analysis (WASA), which can be stated as follows:

$$WASA_i = \frac{1}{W} \sum_{w=1}^{W} Sent_{i-w} \tag{3}$$

Considering that the longer the time interval the smaller the sentiment value, this work also proposes an exponentially weighted sentiment profile, called Exponential Weighted Sentiment Analysis (EWSA), which is:

$$EWSA_i = (1 - \sigma) * EWSA_{i-1} + \sigma * Sent_i, \ 0 < \sigma \leq 1, \ EWSA_0 = 0 \tag{4}$$

Both $WASA_i$ and $EWSA_i$ can be expressed by ei in the problem definition. At this point, we have obtained the sentiment profile time series. After combining it with the stock price time series, we can get the input F for the next stage:

$$F = \text{concate}([X, E]), \ F \in R^{L \times 2} \tag{5}$$

Any one of them is $f_i = \text{concate}([x_i, e_i])$. In our experiments, we analyse the role of sentiment features by removing $e_i$ or switching the type of $e_i$.

## Romember: long-term modelling of sentiment

Firstly, we have opted for the Transformer architecture as the foundation for long-term modelling of sentiment and stock prices. Theoretically, Transformer demonstrates greater dependency correlation capability than LSTM, CNN and other neural networks. However, MultiHeadSelfAttention in the Transformer architecture leads to increased complexity with length $O(L^2)$. In light of this, we initially adopted ProbAttention from Informer to curtail the complexity to $O(L\ln(L))$.

Secondly, it has been observed that there is insufficient distance information available in ProbAttention. Therefore, rotational position encoding has been implemented to enable the model to better comprehend and incorporate positional data to compensate for this insufficiency. This section corresponds to the Romember section on the right-hand side of Fig. 1. The subsequent section will elaborate on the implementation of rotational position encoding.This section corresponds to the Romember section on the right-hand side of Fig. 1. The subsequent section will elaborate on the implementation of rotational position encoding.

## ProbAttention

In the Transformer framework, the input is typically split into three components: query, key, and value. We then proceed to compute the similarity between the query and the key, followed by a weighted aggregation of the associated values using the following formula:

$$A(Q, K, V) = \text{Softmax}\left(\frac{QK^T}{\sqrt{d}}\right)V \tag{6}$$

where $Q \in R^{L_Q \times a}, K \in R^{L_K \times d}, V \in R^{L_V \times a}$. For further elaboration, we let qi, ki, and vi represent Q, K, and V respectively. We calculate the inner product of each element in $a_{ij} = q_i k_j^T$ with all the others, which is the reason why the complexity is $O(L^2)$. The idea of ProbAttention arises from the Informer's temporal prediction work which integrates two additional computation steps into the attenuation process. Firstly, the model randomly choose kj and subsequently compute its inner product with all other elements. Secondly, following the previous step's outcome, the model select $\ln(L_q) \ q_i$ and internally multiply them with all $k_j$ to obtain the attention fractional moments, which produces the attention score matrix. The above two steps' total algorithmic complexity stands at $O(L\text{lin}(L))$.

## Rotational position embedding

We know that Transformer and Informer location information is also necessary to indicate the location of each data point. They can show the location of each data point. However, in attention calculations, they do not reflect the relative information of the two positions, mainly because the dot product of the absolute position encoding after multiple layers of

linear transformations is no longer equal to a function of the difference between the two absolute positions m and n. In ProbAttention, the sum of the filtered values is not necessarily the same as the difference between the two absolute positions. In ProbAttention, the relative position information is more important because the filtered $q_i$ and $k_j$ are not necessarily adjacent to each other. Because $k_j$ and $q_i$ and $q_{i+1}$ respectively to find the inner product, $q_i$ and $q_{i+1}$ are just adjacent to the position. In fact, the middle may be sifted out of the unknown length of the data, but in the view of the $k_j$ they may be adjacent to each other, but also may be inverse order. This undoubtedly leads to a lack of relative positional information and a certain degree of precision loss in the timing task. Therefore, we must attach enough position information to each $q_i$, $k_i$ to indicate the relative position. Rotational Position Encoding (RoPE) is a kind of encoding that can reflect the relative position (*Su et al., 2022*) and has been widely used in generative tasks in natural language processing. Specifically, for any $q_i$, $k_j$, a rotational position encoding $R^d_{\theta,pos}$ is multiplied.

$$rq_i = R^d_{\Theta,i} q_i \quad rk_j = R^d_{\Theta,i} k_j \tag{7}$$

The inner product with the addition of rotational position coding can be expressed as follows:

$$rq_i rk_j^T = \left( R^d_{\Theta,i} q_i \right) \left( R^d_{\Theta,i} k_j \right)^T = q_i R^d_{\Theta,j-i} k_j^T \tag{8}$$

$R^d_{\theta,pos}$ in the above formula accurately represents the relative distance between positions i and j. Its remarkable feature lies in the fact that it comprises rotations of the sine-cosine matrix. This can be expressed as follows:

$$R^d_{\Theta,i} = \begin{pmatrix} \cos i\theta_1 & -\sin i\theta_1 & 0 & 0 & \cdots & 0 & 0 \\ \sin i\theta_1 & \cos i\theta_1 & 0 & 0 & \cdots & 0 & 0 \\ 0 & 0 & \cos i\theta_2 & -\sin i\theta_2 & \cdots & 0 & 0 \\ 0 & 0 & \sin i\theta_2 & \cos i\theta_2 & \cdots & 0 & 0 \\ \vdots & \vdots & \vdots & \vdots & \ddots & \vdots & \vdots \\ 0 & 0 & 0 & 0 & \cdots & \cos i\theta_{d/2} & -\sin i\theta_{d/2} \\ 0 & 0 & 0 & 0 & \cdots & \sin i\theta_{d/2} & \cos i\theta_{d/2} \end{pmatrix}$$

where a is a hyperparameter that can be calculated well in advance by using the following equation:

$$\Theta = \left\{ \theta_i = 10000^{-2(i-1)/d}, i \in [1, 1, \ldots, d/2] \right\} \tag{9}$$

By using rotated positional coding to improve the missing positions in ProbAttention, we can better handle long-term time series and introduce longer-term patterns of change in market sentiment.

| Headlines | Time |
|---|---|
| Samsung Electronics chip output at South Korea plant partly halted due to short blackout | 2020-01-01 |
| Major commercial plane crash deaths worldwide fell by more than 50% in 2019: group | 2020-01-01 |
| More drugmakers hike U.S. prices as new year begins | 2020-01-01 |
| Nissan ex-boss Carlos Ghosn to hold press conference on Jan. 8: lawyer | 2020-01-01 |
| Exclusive: Airbus beats goal with 863 jet deliveries in 2019, ousts Boeing from top spot | 2020-01-01 |
| Surveillance in a leafy enclave, Ghosn's Tokyo life was under strict monitoring | 2020-01-01 |
| Britain's Lloyds Banking Group suffers hours-long online outage | 2020-01-01 |
| Surveillance in a leafy enclave, Ghosn's Tokyo life was under strict monitoring | 2020-01-01 |
| Ghosn flight prompts talk of more curbs in Japan's strict justice system | 2020-01-01 |
| Strike not extended at Lufthansa's Germanwings for now | 2020-01-01 |

**Figure 2  Reuters sample news headlines.**     

## EXPERIMENT

### Data description

The author utilizes two-part experimental data for daily market sentiment analysis. For natural language processing and analysis, the headlines of Reuters news articles on financial, monetary, health, environment, healthcare and pharmaceutical from 30/03/2011 to 17/07/2020 are used. Reuters is a highly regarded international news organization with an extensive history and global reach, recognized for its impartial, objective, and punctual news reporting. News headlines are a timely and important source of market events and information, enabling faster reaction of stock prices. The precise and objective language used can accurately reflect the mood and emotions of market participants, and the extent to which the market responds to specific events. News headlines provide various information concerning the company's performance, industry trends, and policy changes, which can garner an all-inclusive prediction of the stock price.

Alongside this, the daily closing prices of the S&P 500 from 30/03/2011 to 17/07/2020 were utilized in the experimental data. The S&P 500 is a prevalent stock index employed in the US stock market, inclusive of shares from 500 representative US public companies. These companies cover a broad spectrum of industries and sectors. The S&P 500 accurately mirrors the conditions of the entire US market and exhibits the overall trend of the US stock market. The FTSE 100 is an index of the 100 largest companies by market capitalisation on the London Stock Exchange, representing the UK's leading companies across a wide range of sectors such as financial services, energy, consumer goods, *etc.*, and effectively reflecting the state of the UK stock market.

### Data preparations

The first part of the news headline data was obtained from the Reuters website (*Reuter, 2022*) *via* a web crawler. In this study, news headline data was crawled from the financial, monetary, health, environment, healthcare and pharmaceutical sections of Reuters. Figure 2 illustrates the sample of news headlines collected on January 1, 2020.

The following dataset is sourced from the Yahoo Finance API (*Yahoo Finance, 2020*), which provides daily open, close, high, low and adjusted close prices for the S&P 500 and FTSE 100 indices. The Yahoo Finance API comprises numerous interfaces furnished by Yahoo, facilitating the collection of financial and stock market data, including stock quotes, stock news, historical stock data, and financial information.

## Data preprocessing

Since news headlines may contain superfluous information, such as punctuation marks, special characters, repeated words, or suspended words, this experiment aims to eliminate these elements and refine the data of news headlines (*Usmani & Shamsi, 2023*). Meanwhile, the author performs lexical segmentation and annotation, removes inactive words and reduces them to their original or stemmed forms, to standardise news headlines and improve the accuracy of natural language processing.

Meanwhile, we used the roberta.large model (*Liu et al., 2019*), a pre-trained deep learning model for natural language processing, to implement sentiment classification. First, we used roberta.large model's tokeniser to segment text into tokens; the tokeniser tries to split words or text fragments into smaller units until all units can be found in the vocabulary. Then, to identify the start and end of a sentence sequence, roberta.large model used <s> as the start token of the sequence and </s> as the end token of the sequence. After splitting into tokens, these tokens were converted into indexes by looking up the corresponding unique index of each token in roberta.large model's vocabulary. Since the roberta.large model requires that all input sequences have the same length, shorter sequences must be padded and longer sequences must be truncated. In order for the model to know which positions are real tokens and which positions are padded tokens, an attention mask must also be created. For real tokens, the mask value is 1; for padded tokens, the mask value is 0. The above steps convert the raw text into a format that can be processed by the roberta.large model.

## Emotional factor extraction

The author obtained the pre-trained roberta.large model (*Liu et al., 2019*) repository and fine-tuned it by utilizing a curated dataset of news headlines. The training data consisted of 8,000 sentiment sentences which had been manually labeled as either positive, negative, or neutral. Additionally, 1,200 sentences were employed as validation data with a further 1,200 for testing purposes. The model's maximum sequence length was 512, with epochs set at 2 and a learning rate of 0.01.

The author entered the Reuters news headlines into the fine-tuned robera-large algorithm, using EWSA and WASA to determine the respective market sentiment indices. Next, the author merged the obtained sentiment indices with the previously obtained S&P 500 data.

Min-Max Normalization (M-MN) is a commonly used data normalization method, which is implemented by Eq. (10). After merging the investor sentiment index into the stock data, the authors use the max-min normalization to scale the data to [0,1], which makes the range of values of different features uniform.

**Table 1 Indicators with 5, 25, 50 day time lag based on SP500.**

| Model | 7-day time lag | | | 25-day time lag | | | 50-day time lag | | |
|---|---|---|---|---|---|---|---|---|---|
| | MAE | RMSE | MAPE | MAE | RMSE | MAPE | MAE | RMSE | MAPE |
| RNN | 14.041 | 23.227 | 5.072 | 22.59 | 28.856 | 7.981 | 27.456 | 36.387 | 9.485 |
| GRU | 12.225 | 21.424 | 4.439 | 21.73 | 28.024 | 7.119 | 20.701 | 24.277 | 6.878 |
| LSTM | 12.735 | 16.200 | 4.536 | 11.94 | 16.944 | 4.401 | 14.596 | 19.269 | 5.299 |
| Transformer | 10.471 | 14.354 | 6.102 | 9.986 | 12.675 | 6.009 | 7.111 | 9.661 | 4.022 |
| Informer | 3.733 | 4.516 | 1.943 | 4.602 | 6.15 | 2.873 | 4.123 | 5.013 | 2.407 |
| Romember | 3.218 | 4.289 | 1.821 | 3.970 | 4.989 | 2.230 | 3.489 | 4.041 | 2.036 |
| WASA-Romember | 3.027 | 3.929 | 1.577 | 2.987 | 3.740 | 1.710 | 3.073 | 3.701 | 1.661 |
| EWSA-Romember | 2.762 | 3.589 | 1.426 | 2.848 | 3.521 | 1.580 | 2.448 | 2.989 | 1.253 |

**Note:**
The table describes the MAE, MAPE, and RMSE based on SP500 for each subject at time lags of 7, 25, and 50 days, respectively.

$$X_{nor} = (X - X_{min})/(X - X_{max}) \tag{10}$$

The study determined that the hidden states of every layer of Romember were set at 384, while also configuring 300 attention heads. The loss function employed was MSE, with Adam selected as the optimizer at a learning rate of 0.01. The model underwent 100 epochs. Furthermore, the authors included a composite feature comprising investor sentiment index and stock fundamentals data based on initial experiments. Comparative experiments were also carried out with time lags of 7, 25, and 50. Additionally, comparative experiments with time lags of 7, 25, and 50. Table 1 correspond to experimental results with time lags of 7, 25, and 50 respectively. Based on past experimental data, the daily weight given to sentiment was set to 0.94.

## Performance judgement indicators and their rationality analysis

In a financial time series problem such as forecasting stock prices, this study uses Mean Absolute Error (MAE), Root Mean Square Error (RMSE) and Mean Absolute Percentage Error (MAPE) to evaluate the performance of the model. The smaller the value of these indicators, the smaller the difference between the predicted and actual values and the more accurate the model prediction. The selection of indicators and the analysis of their appropriateness are as follows:

Mean Absolute Error (MAE)

Definition: MAE is the average of the absolute value of the difference between the observed value and the true value. It is a very intuitive measure of forecast accuracy.

Rationality: MAE gives equal weight to all error sizes, so it does not overemphasise large errors, which makes it particularly useful in stock price forecasting, where small price fluctuations are very common in the financial markets and MAE can help us to understand how the model is performing on average.

Root Mean Square Error (RMSE)

Definition: RMSE is the square root of the mean of the squares of the differences between the observed and true values. It quantifies the magnitude of the error.

**Table 2 Indicators with 5, 25, 50 day time lag based on FTSE100.**

| Model | 7-day time lag | | | 25-day time lag | | | 50-day time lag | | |
|---|---|---|---|---|---|---|---|---|---|
| | MAE | RMSE | MAPE | MAE | RMSE | MAPE | MAE | RMSE | MAPE |
| CNN | 14.368 | 22.679 | 5.342 | 22.230 | 27.963 | 7.947 | 25.236 | 33.049 | 8.156 |
| RNN | 13.338 | 21.954 | 4.818 | 21.361 | 27.329 | 7.547 | 25.970 | 34.445 | 8.978 |
| GRU | 11.557 | 20.301 | 4.195 | 20.579 | 26.503 | 6.733 | 20.701 | 22.981 | 6.509 |
| LSTM | 12.097 | 15.309 | 4.307 | 11.311 | 16.007 | 4.159 | 14.596 | 18.214 | 5.013 |
| Transformer | 9.497 | 13.574 | 5.776 | 9.436 | 11.998 | 5.689 | 7.111 | 9.136 | 3.801 |
| Informer | 3.546 | 4.268 | 1.840 | 4.352 | 5.822 | 2.716 | 4.123 | 4.742 | 2.285 |
| Romember | 3.057 | 4.054 | 1.726 | 3.762 | 4.715 | 2.108 | 3.489 | 3.829 | 1.934 |
| WASA-Romember | 2.876 | 3.718 | 1.494 | 2.829 | 3.531 | 1.612 | 3.073 | 3.502 | 1.574 |
| EWSA-Romember | 2.621 | 3.396 | 1.352 | 2.696 | 3.332 | 1.498 | 2.448 | 2.830 | 1.189 |

**Note:**
The table describes the MAE, MAPE, and RMSE based on FTSE100 for each subject at time lags of 7, 25, and 50 days, respectively.

Rationality: Compared to MAE, RMSE gives a higher penalty for larger errors (because the error is squared). In stock price forecasting, this means that the RMSE increases significantly if the model has a large prediction error for certain data points. This helps to identify and improve those cases where the model performs poorly at the extremes.

Mean Absolute Percentage Error (MAPE)

Definition: MAPE is the average of the absolute percentage errors between the observations and the true values. It provides an intuitive sense of the magnitude of the error relative to the true value.

Rationality: MAPE is particularly useful in scenarios where the size of the relative error needs to be measured. In stock price forecasting, since the absolute value of the stock price may vary from stock to stock, using MAPE can help us understand the relative accuracy of the model's predictions, which is especially important for comparing the forecasting performance.

## Comparison of experimental results of different models

### Prediction accuracy and robustness

In this study, we comprehensively evaluate the performance of several mainstream deep learning architectures and our three newly developed models under different time delay conditions. The experimental results show that neural network models employing self-attention mechanisms (*e.g.* Transformer, Informer, Romember, WASA-Romember, and EWSA-Romember) significantly outperform traditional recurrent neural networks (*e.g.* RNN, GRU, LSTM) and convolutional neural networks (CNN), demonstrating more accurate prediction capabilities. This result suggests that, under the same experimental conditions, neural network models driven by self-attention mechanisms exhibit superior performance compared to recurrent and convolutional neural networks. It is worth noting that the prediction performance of traditional recurrent and convolutional neural networks shows a decreasing trend with increasing time delay. On the contrary, the prediction performance of transformer-structured neural networks using a self-learning

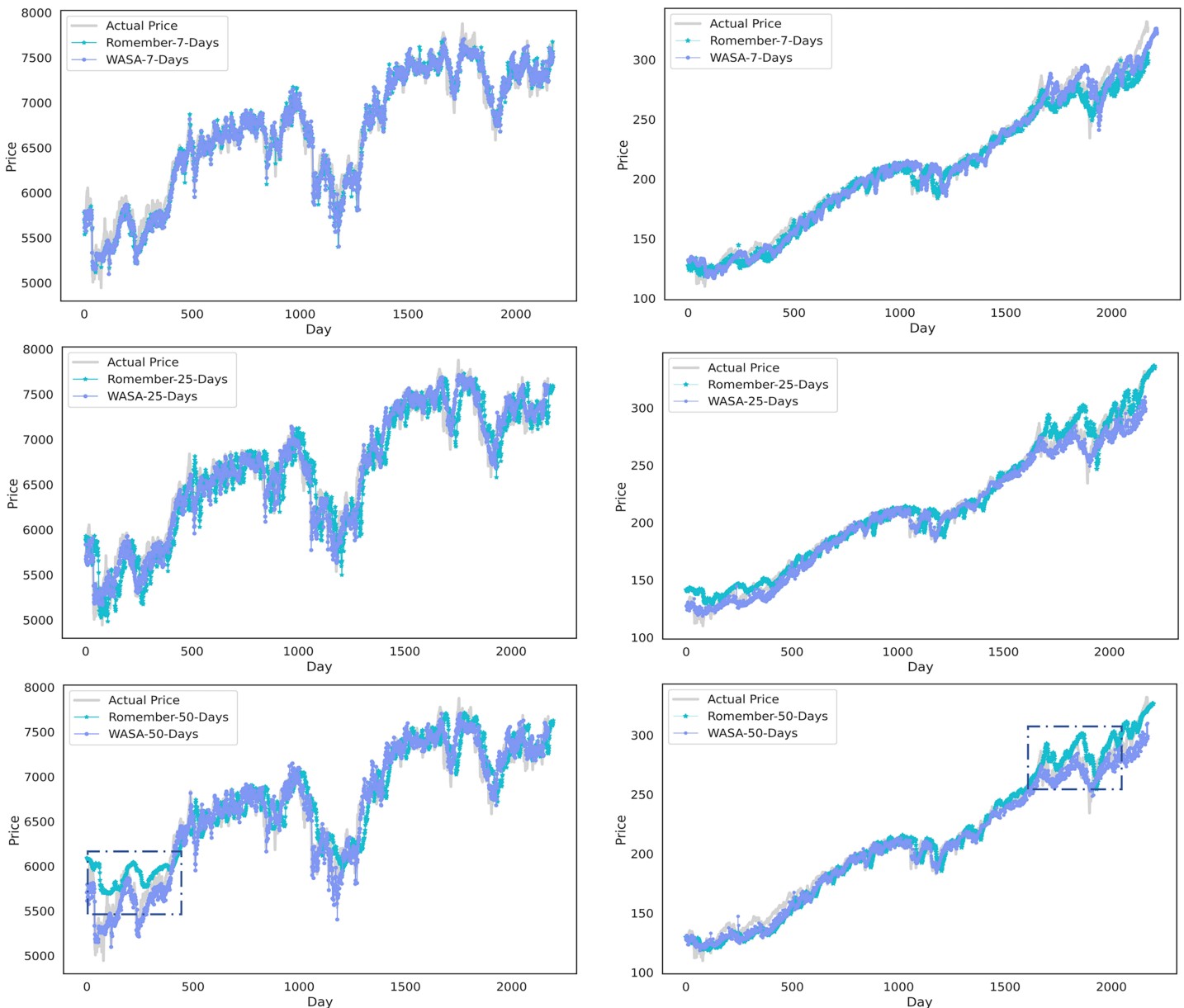

**Figure 3 Comparison of stock price forecast results between Romember and WASA-Romember.** The graph illustrates the comparison of Romember's and WASA-Romember's share price forecasts.     

mechanism increases subsequently. Furthermore, our proposed Romember, WASA-Romember and EWSA-Romember models show similar performance on the SP500 and FTSE100 datasets, demonstrating that these models have good robustness. The results of the experiment are presented in Tables 1 and 2.

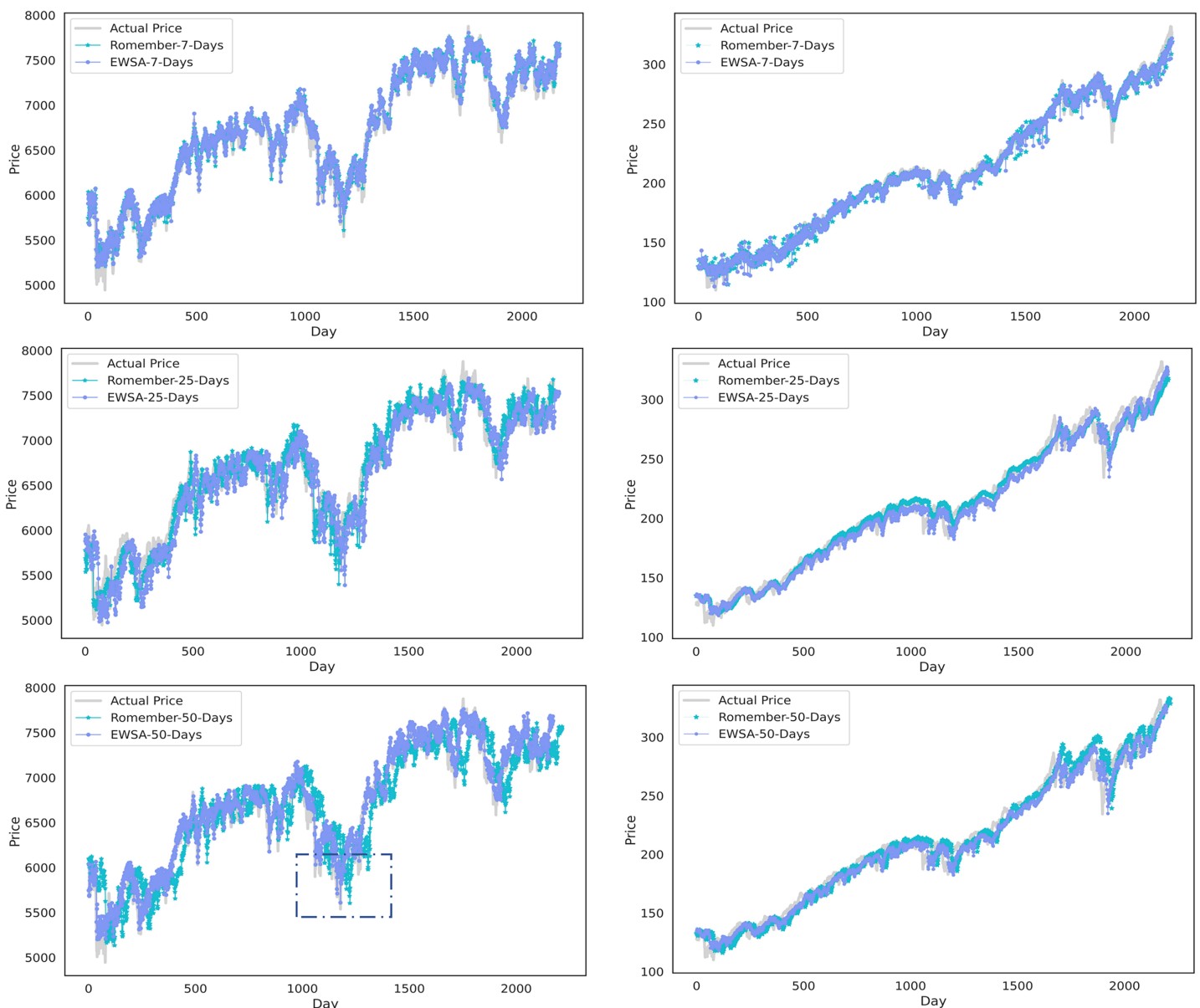

**Figure 4 Comparison of stock price forecast results between Romember and EWSA-Romember.** The graph illustrates the comparison of Romember's and EWSA-Romember's share price forecasts.

### Predictions for Romember with and without the inclusion of emotional features

Figure 3 compares the WASA-Romember forecasts with those of Romember at different lags. The left half of Fig. 3 shows the results based on the FTSE 100 and the right half on the SP 500. In particular, the box in the figure shows that WASA-Romember, which includes sentiment features, outperforms Romember in terms of stock price prediction.

In addition, Fig. 4 compares the results of EWSA-Romember and Romember using different lags. The left half of Fig. 4 shows the results based on the FTSE 100 index and the

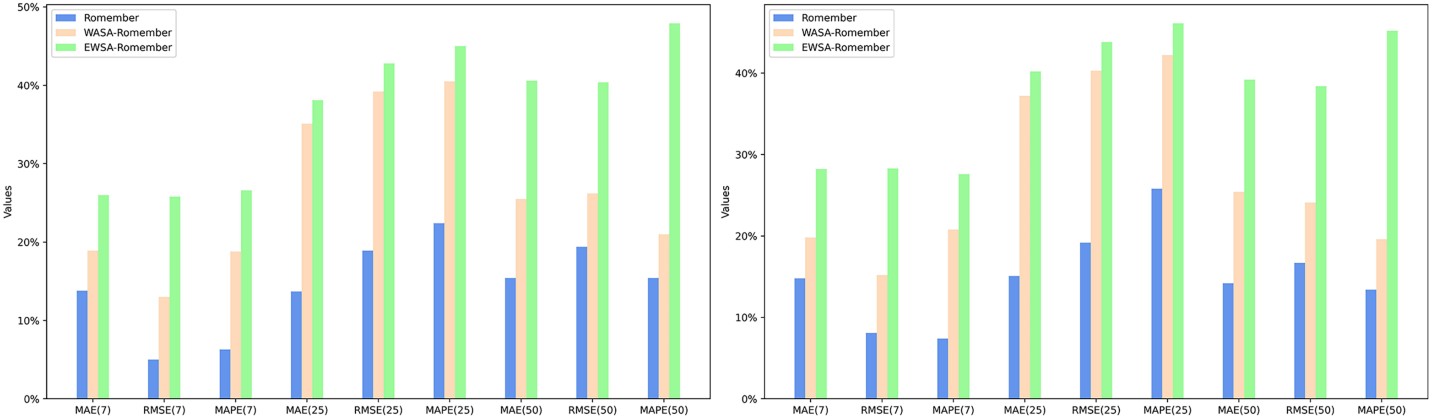

**Figure 5** The percentage improvement in the prediction performance metrics MAE, RMSE, and MAPE over the informer model.

right half shows the results based on the SP 500 index. In particular, the boxed part of the figure shows that EWSA-Romember, which includes a sentiment profile, outperforms Romember in terms of stock price forecasting.

### Comparison of performance between pre-improved model and improved model

*A boost due to rotational position coding*

The experimental results, as shown in Tables 1 and 2, clearly indicate that the Romember model with the rotational position coding strategy significantly outperforms the Informer model in terms of prediction accuracy under the same delay conditions. Further analysis shows that rotational position coding is most effective in improving the predictive performance of the model over a longer time lag of days (25 days), as evidenced by the average rate of improvement in the performance metrics based on mean absolute error (MAE), root mean squared error (RMSE) and mean absolute percentage error (MAPE) for the two datasets: −14.4%, 19.1% and 24.1% respectively. This is particularly true when MAPE is used as the evaluation criterion. As a ratio indicator, MAPE provides a uniform metric that is independent of the stock price range, making it possible to compare model performance across different stocks or markets. It can therefore be concluded that rotational position coding significantly improves the robustness of the model in medium to long term forecasting. The specific experimental results are shown in Fig. 5.

### A boost due to sentiment features

In particular, the inclusion of sentiment features in the Romember model significantly improves its predictive power compared to the version without sentiment features. Furthermore, EWSA-Romember outperforms WASA-Romember with fixed sentiment weights, especially in the long lag days (50 days). When trained on SP500 and FS100 data with a lag of 50 days, EWSA-Romember shows the highest average performance improvement compared to Informer's MAE, RMSE and MAPE metrics, which are 39.9%,

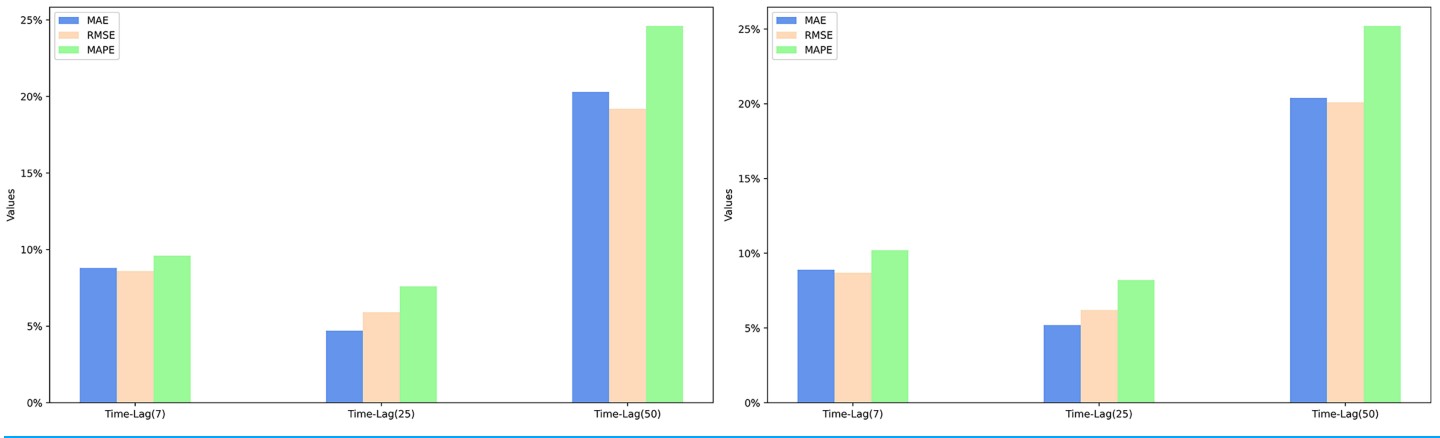

**Figure 6** **Comparison of EWSA and WASA in terms of MAE, RMSE, and MAPE improvements.**

34.4% and 46.6% respectively. Figure 5 shows the performance comparison (in terms of percentage improvement) of Romember, WASA-Romember and EWSA-Romember with the Informer model on the three predictive performance metrics of MAE, RMSE and MAPE compared to Informer, with the results based on the FTSE100 dataset on the left and the results based on the SP500 dataset on the right. The left side is based on the FS100 dataset and the right side is based on the SP500 dataset.

### Comparison of model performance improvement using the EWSA approach compared to the WASA approach

Based on the experimental results, it can be seen that when the lag period is 50 days, EWSA-Romember has the most significant performance improvement in MAE, RMSE and MAPE metrics compared to WASA-Romember, which are 20.4%, 19.6% and 24.9% respectively, which proves the effectiveness of the sentiment smoothing feature. The specific improvement percentages are shown in Fig. 6. Figure 6 illustrates the percentage improvement of the prediction performance metrics MAE, RMSE, MAPE of EWSA-Romember compared to WASA-Romember. The left half of Fig. 6 is the result obtained based on SP500 and the right half is the result obtained based on FTSE100.

## CONCLUSION

In this article, we present the LEET method (Enhanced Time Series Analysis of Long-Term Sentiment Changes), which uses the EWSA and WASA market sentiment index estimation methods and the Romember Long-Order Sentiment Time Series model to accurately and efficiently extract the long-term dependencies between market sentiment and the underlying stock indices. The experimental results show that the LEET method can effectively address the difficulty of most deep learning frameworks in accurately making predictions based on long-term time scales and effectively incorporating market sentiment into models. Meanwhile, by providing a more accurate and comprehensive market sentiment analysis framework, LEET can more effectively reflect the potential market sentiment and trends that are difficult to quantify, and help investors and financial

institutions identify potential market volatility risks and predict market trends more accurately.

## FUTURE WORK

### Model performance

To help investors make more intuitive decisions in the future, we plan to expand our research and development efforts in several key areas:

1. Integration of additional economic indicators

While sentiment analysis and sentiment indices provide valuable insights into market trends, the integration of additional economic indicators can significantly improve the accuracy of a model. These indicators can include macroeconomic data such as GDP growth rates, inflation rates. By analysing the interplay between market sentiment and these economic fundamentals, our models can provide a more complete view of the market and thus predict stock returns with greater accuracy.

2. Extension to global equity markets

Given the interconnected nature of today's global economy, events in one region can have a profound impact on global financial markets. We therefore plan to extend our model to global equity markets, including emerging markets.

3. Real-time data processing

To capture the dynamic nature of financial markets, our future research will focus on implementing real-time data processing. This will allow models to incorporate the latest market news, social media trends and economic reports as they become available. Real-time processing will make our models more sensitive to sudden market changes, providing investors with timely insights to guide their trading decisions.

4. Advanced natural language processing technology

We plan to explore the use of more advanced natural language processing (NLP) technologies that go beyond sentiment analysis. These technologies could include context-aware models capable of understanding the nuances of financial news, such as the difference between the short-term and long-term impact of an event.

5. User-friendly dashboard

We plan to design an intuitive dashboard that presents a holistic view of the market sentiment and trends, integrating the LEET method's insights with additional economic indicators. This dashboard will feature easy-to-understand visualizations, such as sentiment trend graphs, economic indicators charts, and predictive analytics insights. It will enable users to quickly grasp the current market sentiment, understand how it correlates with key economic indicators, and foresee potential market movements.

### Ethical aspect

While the LEET method shows promising results in using emotional analysis for stock market predictions, it is imperative to discuss the ethical and legal implications associated with such methodologies, particularly with respect to privacy and market manipulation.

First of all, the use of emotional analysis in financial markets raises significant privacy concerns. The EWSA and WASA indices, which are fundamental to the LEET

methodology, rely on vast amounts of data extracted from various sources, including social media, news outlets and forums. This data collection process must comply with strict data protection laws, such as the General Data Protection Regulation (GDPR) in the European Union and the California Consumer Privacy Act (CCPA) in the United States. It is crucial to ensure that data is collected and processed transparently, with the explicit consent of individuals, to avoid violating their privacy rights.

In addition, the application of the Romember long order sentiment time series model within the LEET framework requires ethical consideration of how emotional analysis could potentially be exploited for market manipulation. The predictive capabilities of the LEET method, while beneficial for understanding long-term market sentiment, may inadvertently create opportunities for information asymmetry. This asymmetry may enable certain investors to influence market trends based on sentiment analysis, thereby undermining the integrity of financial markets and violating principles of fair trading.

To address the ethical and legal challenges posed by sentiment analysis in stock market forecasting, particularly in terms of data privacy and market manipulation, this study provides a relevant discussion and suggests future improvements:

### Transparent use of data

**1. Data source disclosure:** When using sentiment analysis tools such as the LEET method, clearly list all data sources, including social media platforms, news sites, forums, *etc*., and explain how this data is collected and processed.

**2. User consent for access:** Ensure that explicit user consent is obtained before personal data is collected. This can be done by updating privacy policies, providing easy-to-understand consent forms, *etc*.

### Ethical guidelines for sentiment analysis

**1. Establish ethical guidelines:** Develop a set of ethical guidelines specific to the use of sentiment analysis in stock market forecasting, including but not limited to protecting individual privacy, ensuring data accuracy and avoiding bias.

**2. Avoid market manipulation:** Explicitly prohibit the use of sentiment analysis results for market manipulation or unfair trading practices. This includes preventing trading on the basis of undisclosed information or using sentiment analysis results to mislead other investors.

### Regulatory compliance and oversight

**1. Working with regulators:** Proactively communicating with financial market regulators to ensure that sentiment analysis methods comply with existing market rules and regulations.

**2. Regular audits:** Conduct regular internal or third party audits to ensure that sentiment analysis tools are used in accordance with ethical and legal requirements, particularly with regard to data protection and market fairness.

### Investor education

**1. Transparent risk disclosure:** Clearly disclose the potential risks and uncertainties when investors use forecasting tools based on sentiment analysis.

**2. Promoting rational investment:** Encouraging investors to make comprehensive and rational investment decisions by combining the results of sentiment analysis with other investment information such as fundamental analysis.

### Funding

The authors received financial support from the National Social Science Foundation of China under Grant no. 21BJY033. The funder played a role in the data collection and analysis. The funder played no role in study design, decision to publish, or preparation of the manuscript.

### Grant Disclosures

The following grant information was disclosed by the authors:
National Social Science Foundation of China: 21BJY033.

### Competing Interests

The authors declare that they have no competing interests.

### Author Contributions

- Honglin Liao conceived and designed the experiments, performed the experiments, analyzed the data, performed the computation work, prepared figures and/or tables, authored or reviewed drafts of the article, and approved the final draft.
- Jiacheng Huang analyzed the data, prepared figures and/or tables, authored or reviewed drafts of the article, and approved the final draft.
- Yong Tang analyzed the data, prepared figures and/or tables, authored or reviewed drafts of the article, and approved the final draft.

### Data Availability

The raw data and that Rememeber's Model Architecture model are available in the Supplemental Files.

### Supplemental Information

Supplemental information for this article can be found online at http://dx.doi.org/10.7717/peerj-cs.1969#supplemental-information.

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
