# Peer review of "LEET: stock market forecast with long-term emotional change enhanced temporal model"

_PeerJ Computer Science, doi:10.7717/peerj-cs.1969_

## Round 0.1 · original submission · Major Revisions

Dear authors,


Thank you for submitting your article. The reviewers’ comments are now available. Your article has not been recommended for publication in its current form. However, we encourage you to address the reviewers' concerns and criticisms; particularly regarding experimental design and validity, and resubmit your article once you have updated it accordingly.

Reviewers have requested that you cite specific references. You may add them if you believe they are especially relevant. However, I do not expect you to include these citations, and if you do not include them, this will not influence my decision.

Best wishes,

**Language Note:** The review process has identified that the English language must be improved. PeerJ can provide language editing services - please contact us at copyediting@peerj.com for pricing (be sure to provide your manuscript number and title). Alternatively, you should make your own arrangements to improve the language quality and provide details in your response letter. – PeerJ Staff

·

Basic reporting

This paper proposes the LEET methodology for long-term stock price prediction by effectively extracting dependencies between market sentiment and stock fundamentals. It aims to address limitations of conventional neural networks in capturing long-range dependencies and incorporating sentiment dynamics.

The LEET method utilizes two sentiment index estimation techniques - EWSA and WASA - to derive daily sentiment values from news headlines using pretrained language models. A temporal model named Romember is developed based on the ProbAttention Transformer architecture with rotational position encoding. This enhances positional understanding for sentiments over long periods.

The model is evaluated on Reuters news data and S&P500 prices against standard baselines like RNNs, LSTMs and Transformers, as well as recent techniques like Informer. Results demonstrate Romember outperforms these, and sentiment-integrated versions further improve accuracy versus the base model. EWSA-Romember performs best by assigning decaying sentiment weights.

While presenting an interesting first study, some key aspects need further elaboration to meet top-tier standards. These include more comprehensive benchmarking, rigorous statistical validation of gains, and analyses of limitations and practical impacts. Also, sensitivities to external factors and hyperparameter impacts warrant exploration. Nonetheless, the work provides a valuable starting point for incorporating long-term market psychology into predictive stock modeling.

Experimental design

See Additional comments

Validity of the findings

See Additional comments

Additional comments

1.The literature review could be more comprehensive by discussing additional recent works that have incorporated sentiment analysis or self-attention mechanisms into stock prediction models.

2.More details on the data collection and preprocessing steps could strengthen reproducibility, such as the specific news sources and dates.

3.Comparative experiments against more baseline models would further validate the proposed method's effectiveness.

4.Quantitative analysis of the sentiment extraction components' contributions is lacking.

5.The rotational position encoding merits deeper exploration of its effects versus absolute encodings.

6.More discussion of practical application potential and real-world trading implications would boost impact.

7.As stock predictions depend on various external factors, addressing model robustness under changing market conditions could strengthen the findings.

8.The cited literature is not new enough. For example, you can refer to some new integrated learning methods:
(1)Gong, P., Liu, J., Zhang, X., & Li, X. (2023, June). A Multi-Stage Hierarchical Relational Graph Neural Network for Multimodal Sentiment Analysis. In ICASSP 2023-2023 IEEE International Conference on Acoustics, Speech and Signal Processing (ICASSP) (pp. 1-5). IEEE.
(2)Xiao, L., Wu, X., Yang, S., Xu, J., Zhou, J., & He, L. (2023). Cross-modal fine-grained alignment and fusion network for multimodal aspect-based sentiment analysis. Information Processing & Management, 60(6), 103508.

9.Readability and language quality could be improved in some sections to enhance comprehension for a global readership.

Reviewer 2 ·

Basic reporting

The study presented in the manuscript focuses on enhancing stock market prediction by integrating long-term market sentiment analysis with advanced deep learning techniques. The primary contribution lies in the development of the LEET (Long-term Sentiment Change Enhanced Temporal Analysis) method, which incorporates two novel market sentiment index estimation methods: EWSA (Exponential Weighted Sentiment Analysis) and WASA (Weighted Average Sentiment Analysis). Additionally, the study employs a Transformer architecture based on ProbAttention with rotational position encoding to capture long-term emotional trends more effectively.
To further improve the quality of this study, the authors could consider the following suggestions:
1. Strengthen the experimental validation of the LEET method using diverse datasets, including different stock markets and periods, to demonstrate the robustness and adaptability of the model.
2. Provide a more detailed comparative analysis with existing models, highlighting the advantages and improvements the LEET method offers in various scenarios.
3. Enhance the sentiment analysis component by exploring more complex natural language processing techniques and considering diverse sources of sentiment data, such as social media and financial news.
4. Address how the model accounts for sudden, unexpected market changes and demonstrate its effectiveness in such scenarios.
5. Consider developing a user-friendly interface that investors can utilize, providing actionable insights and predictions in an easily interpretable format.
6. Discuss any ethical and legal implications of sentiment analysis in stock market prediction, especially regarding data privacy and market manipulation.
7. Outline potential future enhancements to the model, such as integrating additional economic indicators or expanding to global stock markets.
8. Ensure that the manuscript is clear, well-organized, and free from technical jargon, making it accessible to a broader audience, including practitioners in the finance sector.
9. To enhance the depth and relevance of your study, it's recommended to explore, integrate, and acknowledge recent scholarly works in your field. Including insights and findings from these contemporary publications can significantly enrich your research's contextual framework and theoretical underpinnings.
i. Luo, J., Zhuo, W., & Xu, B. (2023). A Deep Neural Network-based Assistive Decision Method for Financial Risk Prediction in Carbon Trading Market. Journal of Circuits, Systems and Computers. doi: 10.1142/S0218126624501536
ii. Liu, B., Li, M., Ji, Z., Li, H., & Luo, J. (2024). Intelligent Productivity Transformation: Corporate Market Demand Forecasting With the Aid of an AI Virtual Assistant. Journal of Organizational and End User Computing (JOEUC), 36(1), 1-27. http://doi.org/10.4018/JOEUC.336284
iii. He, C., Huang, K., Lin, J., Wang, T., & Zhang, Z. (2023). Explain systemic risk of commodity futures market by dynamic network. International Review of Financial Analysis, 88, 102658. doi: https://doi.org/10.1016/j.irfa.2023.102658
iv. Ding, K., Choo, W. C., Ng, K. Y., & Zhang, Q. (2023). Exploring changes in guest preferences for Airbnb accommodation with different levels of sharing and prices: Using structural topic model. Frontiers in psychology, 14, 1120845. doi: 10.3389/fpsyg.2023.1120845
v. Li, X., & Sun, Y. (2021). Application of RBF neural network optimal segmentation algorithm in credit rating. Neural Computing and Applications, 33(14), 8227-8235. doi: 10.1007/s00521-020-04958-9
vi. Hao, S., Jiali, P., Xiaomin, Z., Xiaoqin, W., Lina, L., Xin, Q.,... Qin, L. (2023). Group identity modulates bidding behavior in repeated lottery contest: neural signatures from event-related potentials and electroencephalography oscillations. Frontiers in Neuroscience, 17. doi: 10.3389/fnins.2023.1184601
vii. Wang, K., Hu, Y., Zhou, J., & Hu, F. (2023). Fintech, Financial Constraints and OFDI: Evidence from China. Global Economic Review, 52(4), 326-345. doi: 10.1080/1226508X.2023.2283878
viii. Wu, H., Jin, S., & Yue, W. (2022). Pricing Policy for a Dynamic Spectrum Allocation Scheme with Batch Requests and Impatient Packets in Cognitive Radio Networks. Journal of Systems Science and Systems Engineering, 31(2), 133-149. doi: 10.1007/s11518-022-5521-0
ix. Li, Z., Zhou, X., & Huang, S. (2021). Managing skill certification in online outsourcing platforms: A perspective of buyer-determined reverse auctions. International Journal of Production Economics, 238, 108166. doi: https://doi.org/10.1016/j.ijpe.2021.108166
x. Huang, C., Han, Z., Li, M., Wang, X., & Zhao, W. (2021). Sentiment evolution with interaction levels in blended learning environments: Using learning analytics and epistemic network analysis. Australasian Journal of Educational Technology, 37(2), 81-95. doi: 10.14742/ajet.6749

Experimental design

Nil

Validity of the findings

nil

Additional comments

nil

---

## Round 0.2 · accepted · Accept

Dear authors,

Thank you for the revision and for clearly addressing all the reviewers' comments. I confirm that the paper is improved and addresses the concerns of the reviewers. Your paper is now acceptable for publication in light of this revision.

Best wishes,

·

Basic reporting

Accept

Experimental design

Accept

Validity of the findings

Accept

Additional comments

Accept

Reviewer 2 ·

Basic reporting

The authors have addressed all the concerns raised and commented on by me. The manuscript is now suitable for publication, so I recommend that it be accepted for publication.

Experimental design

No comment

Validity of the findings

No comment

Additional comments

None